# Dielectric, AC Conductivity, and DC Conductivity Behaviours of Sr_2_CaTeO_6_ Double Perovskite

**DOI:** 10.3390/ma15124363

**Published:** 2022-06-20

**Authors:** Muhammad Zharfan Halizan, Zakiah Mohamed

**Affiliations:** Faculty of Applied Sciences, Universiti Teknologi MARA, Shah Alam 40450, Selangor, Malaysia; halizanzharfan@yahoo.com

**Keywords:** double perovskites, structural properties, optical properties, dielectric properties

## Abstract

Relatively new double perovskite material, Sr_2_CaTeO_6_, has been prepared through conventional solid-state procedures. Structural, dielectric, and optical characteristics of this exquisite solid-state material were analysed in this study. The single-phase monoclinic *P2_1_/n* structure of this prepared compound was well correlated with the literature review. Good distribution of grain sizes and shapes was observed in the morphological study of this compound. The discussions on its optical and dielectric properties are included in this manuscript. High dielectric real permittivity, low dielectric loss, and good capacitance over a range of temperatures possessed by this compound, as shown in dielectric and electrical modulus studies, indicated good potential values for capacitor applications. The *R_o_*(*R_g_Q_g_*)(*R_gb_Q_gb_*) circuit fitted well with the impedance and electrical modulus plot of the compound. Its relatively high electrical DC conductivity in grain at high frequencies and its increasing value with the temperature are typical of a semiconductor behaviour. This behaviour might be attributed to the presence of minor oxygen vacancies within its lattice structure and provides a long-range conduction mechanism. A small difference between activation energy and *E_a_* of DC conductivity indicates that the same charge carriers were involved in both grains and the grain boundaries’ long-range conduction. The electrical AC conductivity of this compound was found to contribute to the dielectric loss in grain structure and can be related to Jonscher’s power law. The presence of polarons in this compound was exhibited by non-overlapping small polaron tunnelling (NSPT) and overlapping large polaron tunnelling (OLPT) conduction mechanisms over a range of temperatures. Wide optical band gap and *E_opt_* in the range of 2.6 eV to 3.6 eV were determined by using an indirect and direct allowed mechanism of electrons transitions. These values supported the efficient semiconducting behaviour of the grain in this material and are suitable for applications in the semiconductor industry.

## 1. Introduction

In the last decades, the introduction of lead halide perovskites was well accepted by global researchers as one of the important discoveries in scientific studies for green technology development [1]. Additionally, many other perovskites have been introduced and studied to enhance their basic properties or even induce new properties. For example, double perovskites were introduced in the configuration of *A_2_BB*′*O_6_*, where A or A′ consisted of alkaline earth or rare earth metals and B and B′ consisted of transition metals. With the rock-salt-like structure of BO_6_ and B′O_6_ units in the crystal, various substitutions or doping can be performed in the B-site of the perovskite to elevate the compounds’ properties such as magnetocaloric in Sm_2_NiMnO_6_ [2], magnetodielectric in Eu_2_NiMnO_6_ [3], and optical properties in Ba_2_ZnWO_6_ or Sr_2_NiWO_6_ [4,5,6]. The rise of the tellurium-based double perovskites with studies on structural and magnetic such as Sr_2_Co_1−x_Mg_x_TeO_6_ and Sr_2_MnTeO_6_ [7,8] contributed to these materials fields in the last few years.

In addition to magnetic properties, tellurium-based oxides were reported to exhibit low dielectric loss and can be synthesised at low sintering temperatures [9]. The dielectric properties in microwave frequencies for Sr_2_ZnTeO_6_ with a tetragonal *I4/m* structure were studied for possible resonator applications. The obtained dielectric real permittivity value lies between 10 and 20 with a good quality factor [10]. Meanwhile, the dielectric investigation in radio frequency for Pb_2_MnTeO_6_ with monoclinic *I2/m* symmetry was reported. It showed a good dielectric real permittivity (~90.0 at 900 kHz at ambient temperature) accompanied by low dielectric loss (less than 1.0 at 900 kHz at ambient temperature), which could be attributed to its intrinsic electrical conductivity [11].

In a previous study, dielectric behaviour in microwave frequencies of *A*_2_MgTeO_6_ (*A* = Ba, Sr, Ca) double perovskites was studied. Sr_2_MgTeO_6_ were crystallised in a tetragonal *I4/m* structure and showed the highest *ɛ′* value among these materials with a value of 14.3. Meanwhile, Ca_2_MgTeO_6_ with monoclinic *P2_1_/n* structure registered the second-highest *ɛ′* value among these materials with a value of 13.2 [12]. These interesting dielectric results in microwave frequency ranges could be associated with values of tolerance factor or densification of each compound [12]. With both Sr_2_MgTeO_6_ and Ca_2_MgTeO_6_ recorded the best dielectric performance, the combination of Sr^2+^ and Ca^2+^ cations in a double perovskite could be expected to produce a compound with a good dielectric performance. In 2005, Sr_2_CaTeO_6_, which applied both of these cations, was discovered and was found to crystallise in monoclinic *P2_1_/n* structure [13]. Most of the work on this compound was focused on structural or optical properties [14,15]. The dielectric behaviour of Sr_2_CaTeO_6_ has not yet been investigated and reported. In addition, the optical band gap, *E_opt_*, of this particular compound was reported to be at 2.75 eV, which can be associated with semiconducting character [15]. Hence, it is interesting to conduct a study on the dielectric properties of Sr_2_CaTeO_6_ and understand its relation to its semiconducting character. Pertained to that, it is also imperative to study the modulus and electrical conductivity properties of this material to determine the possible DC and AC conduction mechanisms that can contribute to any dielectric loss alongside the plausibility of dielectric relaxation existence in this compound.

Therein, Sr_2_CaTeO_6_ was prepared, and its structure, dielectric, and electrical properties were studied with supporting optical properties to further understand the behaviour of this double perovskite material for future references in any related applications.

## 2. Materials and Methods

The Sr_2_CaTeO_6_ compound was prepared by using a conventional solid-state reaction route. Materials with high purity (≥99.99%), including calcium oxide (CaO), tellurium dioxide (TeO_2_), and strontium carbonate (SrCO_3_), were purchased from Sigma-Aldrich (St. Louis, MI, USA) and utilised as precursor powders. Stoichiometry was calculated before mixing all the required chemical precursors. The mixed chemical was ground for 1 h in an agate mortar. The calcinating process was performed at 800 °C in ambient air for 24 h in CWF 11/5 furnace (Carbolite Gero, Hope, UK) to remove the chemical ores within the sample. In order to obtain the pellet form, samples were pressed under 4.5 T of pressure using an Atlas 15 T hydraulic press (Specac, Orpington, UK). The sintering process then proceeded at 1000 °C for 24 h duration in order to form the required compound. A heating rate of 15 °C/min and a cooling rate of 1 °C/min was chosen for both calcinating and sintering procedures to maintain the calculated stoichiometry [16]. X-ray diffraction (XRD) characterisation was carried out in the range of 10° to 80° using Xpert PRO MPD diffractometer (PANanalytical, Almelo, The Netherlands), which was equipped with Cu K_α_ radiation that has 1.5418 Å of wavelength to determine the purity of the prepared compound. In order to refine the XRD result, the data were refined by employing the graphical user interface (EXPGUI) [17,18]. The density of the sample was calculated using Archimedes’ formula. Scanning electron microscope (SEM) and energy dispersive X-ray (EDX) characterisations were conducted in order to determine the morphology structure and constituent elements of the sample, respectively. This work was conducted using LEO model 982 Gemini equipment (Graz, Austria). Fourier transform infrared (FTIR) characterisation was employed to collect the infrared transmittance spectra of the samples within the wavenumber range of 400 to 1500 cm^−1^ using Drift Nicolet 6700 spectrometer (Thermo Fisher, Waltham, MA, USA). In order to investigate the dielectric properties of the compound, electrochemical impedance spectroscopy (EIS) characterisation was conducted within the 50 Hz to 1 MHz of frequencies with 298 K to 343 K of temperature. The sample was held in sandwich geometry by the electrodes while being connected to LCR 3532–50 HiTester analyser (Hioki, Nagano, Japan). Finally, the ultraviolet-visible light (UV–vis) characterisation within the range of 200 to 1000 nm was applied to study the optical spectrum of the sample using a Lambda 750 spectrometer (Perkin Elmer, Waltham, MA, USA).

## 3. Results and Discussion

Figure 1a,b show the XRD plot of Sr_2_CaTeO_6_ and the refined data by the Rietveld refinement method. From refinement data, this compound formed in a single phase without any secondary phase(s) present with reliabilities (*χ*^2^) of 1.15, which shows the very reliable refinement result as per the literature review. The single phase formed in a monoclinic structure of *P2_1_/n* symmetry. The obtained unit cell volume (*V*) for the compound was 278.5 Å^3^, with the lattice constants of 5.804, 5.838, and 8.219 Å, for *a*, *b,* and *c*, respectively. Table 1 shows the complete parameters that were obtained from the refinement. The refined structure parameters were comparable to the other report [14]. Figure 2 shows a refined structure of Sr_2_CaTeO_6_ from the *bc* plane. It is clear that A-site cations (Sr^2+^) are placed between B-site (Ca–O) and (Te–O) octahedral to strengthen the octahedral structure.

From Table 1, there are different bond lengths existing within the B-site octahedral structure. A longer Ca–O bond compared to Te–O bond was attributed to the larger size and lower electronegativity Ca^2+^ compared to Te^6+^. Additionally, a stronger hexavalent hold of the Te–O bond could be the other factor that contributed to the aforementioned difference. The different bond lengths for each Te–O_1_, Te–O_2_, and Te–O_3_ and Ca–O_1_, Ca–O_2_, and Ca–O_3_ bonds indicate distortion in the octahedral shape and deviation from the equivalent bond lengths of the octahedral structure. Ca–O octahedral formed in elongate–like structure with the longest bond at Ca–O_3_. Moreover, Te–O octahedral formed the compressed-like structure since the shortest bond was Te–O_3_. The angle of 157.2° between Ca–O–Te can be referred to as the inclination between different octahedral structures. The deviation of the angle from 180° supported that structural distortion occurred in this compound. This value of bond angles indicates a higher inclination than other reported tellurium-based double perovskites [19,20]. The smaller value can be deduced due to the larger size of Ca^2+^ at the B-site location in this compound and thus, decreasing the structural integrity. From Figure 2, the different locations of Te^6+^ in both compounds indicate that the B-site cations could fill any suitable locations. Nevertheless, the large difference in charge between Te^6+^/Ca^2+^ cations ensured that these cations’ locations were arranged periodically. Since there is a single type of A-site cations, hence, the formation of a layered arrangement of the cations can be withdrawn. Thus, any anion vacancies to accommodate this arrangement can be dismissed [21]. However, since our solid-state method required the sintering at a temperature higher than 1000 °C, the evaporation of CaO chemical could take place [22]. The oxygen vacancy in this sample was determined approximately to *δ* = 0.05 as per refinement.

In order to calculate the tolerance factor of the sample, *τ*, we applied Equation (1) [3]:(1)$$\tau =\frac{R_{A}{+R}_{O}}{\sqrt{2}(\frac{R_{B}{+R}_{B^{\prime }}}{2}{+R}_{O\;})}$$
where *R_A_* = radii of the A-site atom, *R_B_* and *R_B′_* = the radii of the B-site atoms, and *R_O_* = the radius of the oxygen atom in Å.

The calculated tolerance factor (*τ*) of the compound was 0.921, which shows a deviation from the cubic structure. This value confirmed the distortion occurrence between A-site and B-site and hence, distorted the shape of octahedral structures in this material. The size of cations used for the calculation were 1.44 Å (Sr^2+^) with coordination number (CN) of 12 and 1.00 Å (Ca^2+^), 0.56 Å (Te^6+^) and 1.40 Å (O^2−^) with CN of 6 [23]. The calculated tilting angle for the compound can be calculated by using Equation (2) [24]:(2)$$\phi =\frac{180-\theta }{2}$$
where *θ* is the average angle for the (Ca–O–Te) bond. This value indicated the distortion and supported the inclination angle between octahedral structures within the compound. Meanwhile, the crystallite size (*D*) was calculated by using the Scherrer formula [25]:(3)$$D=\frac{K\lambda }{\beta \left(\theta \right)cos\theta }$$
where *K* is the constant value, which is 0.9; *β* is the full width at half maximum (FWHM); *θ* is the angle of XRD peaks; and *λ* is the wavelength of XRD beam in nm [26,27]. Both the calculated tilting angle and crystallite size of this compound were 11.40° and 10.71 nm, respectively.

**Table 1 materials-15-04363-t001:** Obtained space symmetry, parameters of lattice, *V*, angles of bonds, lengths of bond, and fit goodness from Rietveld refinement with calculated *ϕ*, *τ*, and *D* in Sr_2_CaTeO_6_.

Lattice Parameters		Bond Lengths (Ǻ)	
Space Symmetry	*P2_1_/n*		
***a* (Å)**	5.804 (2)	**Ca-O_1_ (x 2)**	2.058 (9)
***B* (Å)**	5.838 (7)	**Ca-O_2_ (x 2)**	2.199 (8)
***c* (Å)**	8.219 (4)	**Ca-O_3_ (x 2)**	2.258 (12)
** *α* **	90.00°	**Ave. < Ca-O >**	2.172 (10)
** *ß* **	90.21°	**Te-O_1_ (x 2)**	1.992 (8)
** *γ* **	90.00°	**Te-O_2_ (x 2)**	2.194 (8)
**Unit cell volume, *V* (Å^3^)**		**Te-O_3_ (x 2)**	1.899 (10)
278.5		**Ave. < Te-O >**	2.028 (9)
**Tilting angle, *Φ***		**Ave. < Sr-Sr >**	4.119 (11)
11.40°		**Bond angles (°)**	
**Fit Goodness**			
** *χ* ^2^ **	1.150		
***R_p_* (%)**	4.9	**Ca-O_1_-Te**	158.1 (1)
***R_wp_* (%)**	6.4	**Ca-O_2_-Te**	150.9 (1)
**Crystallite size, *D***		**Ca-O_3_-Te**	162.5 (1)
10.71 nm		**Aver. < Ca-O-Te >**	157.2 (1)

Figure 3 illustrates the FTIR spectrum of the Sr_2_CaTeO_6_ compound. Some important peaks were present in this spectrum. Peaks at 554 cm^−1^ correlated with Ca–O bond stretching vibrations in octahedral structures [28]. The emergence of medium peaks at 491, 508, 519 and 536 cm^−1^ can be detected and be assigned to antisymmetric stretching vibrations (*v*_1_) of Te–O bonds in the octahedral structures. At the same time, strong peaks at 555, 579, 602, 613, 638, 657, 680, 696, 720, 748, and 763 cm^−1^ were clear and can be related to the symmetric stretching vibration (*v*_2_) of the Te–O bonds [29,30,31,32,33,34]. These peaks’ existence proved the formation of Ca/Te–O bonds in this sample.

Figure 4a exhibits the morphology of the Sr_2_CaTeO_6_ compound. The formation of agglomerated grains with almost the same shape and size distribution as most grains was clear. The agglomerated particles in this compound were most probably due to the fast nucleation rate of grains, which was caused by high temperatures of calcination and sintering. Thus, this caused the grains to become accumulated since there was not enough duration for the grains to separate well from each other [35]. The formation of grains can be confirmed in Figure 4b, where the shape of the top cross-section of a grain (marked with a yellow circle in Figure 4a) formed a hill shape. The obtained grain size distributions are displayed in Figure 4c. The measured distribution grain sizes were within the range of 0.7–2.5 μm, with the majority of the grains’ sizes between 1.0 and 1.5 μm, while there were fewer grains with sizes greater than 2.0 μm. These sizes were relatively larger compared to other tellurium-based double perovskites [36,37]. EDX graph in Figure 4a exhibits constituted elements in this sample, which confirms that these compounds contain elements of the prepared raw material composition, except Au, which originates from the coating in sample preparation for SEM characterisation. The sample’s density is 5.755 g cm^−3^, as calculated by the Archimedes method.

In order to obtain the basic idea of electrical conductivity in this compound, we utilised the Nyquist plot for this purpose. Figure 5a,b exhibit the raw data and fittings of the Nyquist plot of Sr_2_CaTeO_6_ at both the full range of frequencies and high frequencies range. Figure 5c shows the resistance obtained for two different regions, which are grain boundaries and grains in the compound. Figure 5d exhibits the variation in DC conductivity in both grain boundaries and grain. Both of these plots show a decrease in resistance and DC conductivity against temperature and inverse of temperature, respectively.

From Figure 5a, the almost straight line observed has proved that the resistive/capacitive behaviour of this compound is obvious. In addition, there is a small curve in the high frequencies range that indicates the presence of two different regions or mechanisms in this compound. In order to obtain the idea of basic behaviour and the resistance of grain boundaries and grains, the fitting and extrapolation of Nyquist data from low to high frequencies regions and at high frequencies only were carried out by using the Zview software. Based on these fittings, the Nyquist plot of Sr_2_CaTeO_6 is_ suited to the grain and grain boundaries system and includes a constant phase element (CPE) and a resistor at each region. The presence of CPE indicates the non-ideal capacitive behaviour and possesses small electrical conductivity values. From the extrapolation from the fittings, the non-perfect semicircle shape at each fitting implies that the non-Debye impedance occurred within the compound. In order to obtain the resistance values at both grain boundaries and grains, we determined the *x*-axis intercept from the extrapolation data. In Figure 5c, it is clear that the values of grain boundaries and grain resistances decrease as the temperature is elevated.

The decreasing value of resistances in Sr_2_CaTeO_6_ from Figure 5c indicates the negative temperature coefficient resistance (NTCR) behaviour, which is a characteristic typically possessed by a semiconductor. The value of grain resistances is much smaller by a factor of two compared to grain boundaries, which indicates the resistive property of grain boundaries that enclose the more conductive grains. In order to calculate the values of DC conductivity of this compound, Equation (4) was applied [38]:(4)$$\sigma =\frac{t}{RA}$$
where *t* = thickness of the sample, which is 2.05 mm; *R* = resistance of grain boundaries/grain; and *A* = area of electrodes attached to sample, which is 0.19 cm^2^. From Figure 5d, the DC conductivity at various temperatures obeyed the Arrhenius equation, as in Equation (5) [38]:(5)σdc=σoe(−EakBT)
where *E_a_* = activation energy in eV, *k_B_* = Boltzmann constant, and *T* = temperature in K. Hence, the obtained values of activation energy for grain boundaries and grain were 92 and 81 meV, respectively. The small difference between both energy values implies that the same charge carriers were involved for both grain boundaries and grain DC conduction. Moreover, these relatively small values of energy indicate the presence of free electrons in the grain or crystal lattice structure and near grain boundary regions. The presence of these free electrons most probably originated from the singly or doubly ionised oxygen vacancies, which could occur at high temperatures amidst the sintering process. The formed single or double ionised vacancies then produced the electrons by Equations (6) and (7) [39]:(6)$$O_{0}^{x}={\dot{V}}_{O}(int)+e^{\prime }+\frac{1}{2}\;O_{2}(g)$$
(7)$$O_{0}^{x}={\ddot{V}}_{O}(int)+2e^{\prime }+\frac{1}{2}\;O_{2}(g)$$
where ${\dot{V}}_{O}$ and ${\ddot{V}}_{O}$ = single and double ionised oxygen vacancies, respectively.

In order to understand the ability of a compound to store static charge, it is crucial to undergo a dielectric study of a sample. The equation for the complex dielectric permittivity is given as in Equation (8) [40]:(8)$$\varepsilon *(\omega )=\frac{Z^{\prime \prime }}{{{(Z}^{\prime })}^{2}{+(Z}^{\prime \prime }{)}^{2}}+i\frac{Z^{\prime }}{{{(Z}^{\prime })}^{2}{+(Z}^{\prime \prime }{)}^{2}}=\varepsilon^{\prime }+i\varepsilon^{\prime \prime }$$

Figure 6a exhibits the plot of dielectric real permittivity, *ε*′, with respect to frequencies from 50 Hz to 1 MHz in Sr_2_CaTeO_6_ compounds at different temperatures. The variation in *ε*′ at each temperature shows the same shape and trend. At frequencies lower than 100 kHz, the huge drop in *ε*′ can be observed before the slope of the drop decreases at higher frequencies until 1 MHz. At minimum and maximum frequencies, values of *ε*′ in ambient temperature were 6800 and 3800, respectively. As the temperature increased, the *ε*′ values increased accordingly. Figure 6b illustrates the variation in *ε*′ vs. temperatures at specific frequencies. Dispersion of *ε*′ at higher temperature and the enhancement of *ε*′ at different temperatures shows the temperature dependence behaviour of dielectric property in this compound. Moreover, Figure 7c shows the tangent loss, *Tan δ*, of the sample across frequencies at selected temperatures. The values of *Tan δ* registered an increase in temperature with the presence of a peak at initial frequencies and around 100 kHz. At minimum frequency, the value of the loss was 0.25, while at maximum frequency, the loss registered at 0.05 at ambient temperature. At the second peak, the value of the loss was around 0.2 at the ambient temperature.

From Figure 6a, the high values of *ε*′ were probably due to the space charge polarization, where this effect was originated from the movement of charge carriers (heavy electrical dipoles) from oxygen vacancies defects within the grains or between the semiconducting grain and the more resistive grain boundaries, which then caused the heavy electrical dipoles to accumulate. Generally, all types of electrical polarisation, including dipolar, atomic, and electronics polarisations, were involved at initial frequencies and hence, contributed to the large total of dielectric real permittivity value in the compound. The large *ε*′ values in this compound are much larger than reported tellurium based double perovskites with *AA*′*BteO_6_* configuration such as SrLaLiTeO_6_, SrBiLiTeO_6_, and BaLaNaTeO_6_ but with slightly larger dielectric loss [11,36,37]. The difference in grain size between these compounds could provide the reasoning for the *ε*′ value’s difference. Nevertheless, the sudden reduction in *ε*′ at low frequencies was caused by the impotency of the aforementioned charges to follow the alternation of the electric field. Hence, only light electrical dipoles were able to complete the alternation with the external field until the maximum frequency. From Figure 6b, the enhancement of *ε*′ at every temperature indicates the non-relaxor type of polarisation in the compound and the absence of ferroelectricity transition characteristics within the studied temperature range. From both figures, the increase in polarisation with temperature indicated the conservation of total energy from thermal to kinetic energy, which aided dipoles polarisations. In this compound, the thermally activated polarisation at low frequencies can be related to space charge polarisation, while less temperature dependency of polarisation at higher frequencies can be related to lighter dipole polarisation. On the other hand, the stable increase in *ε*′ over a wide range of frequencies at every temperature increment may suggest good polarisation, especially from ionic/atomic polarisation.

From Figure 6c, the high values of *Tan δ* at initial frequencies corresponded to the drop in *ε*′ in the same frequency range as in Figure 6a due to the movement of charge carriers within the grains or between the grain boundaries and grains. This movement would then cause the leakage current flow in the form of DC conduction during the alternation of the external field. This deduction is supported by the value (−1) of the gradient in log *ε*″ vs. log *f* plot in Figure 6c. In addition, the inability of these electrical charges to effectively abide by the external field alternation could be a factor in assisting the high value of the loss. At low frequencies, the increased value of loss as temperature increases implied the increase in dynamic of the charge carriers and contributed to the enhancement of DC conductivity. The existence of peaks at higher frequencies at each temperature and their shifting towards the higher temperatures implied the presence of relaxation peaks at the respective frequencies. The presence of these peaks indicated the impotence of the charges to follow the alternation of the field due to AC conductivity or heat loss. The increase in the peak loss with temperature, however, indicated the increase in carriers’ movements, which strongly implies the presence of AC conductivity at high frequencies.

In order to obtain the full idea of the electrical contributions in the sample, the electrical modulus study can be carried out for this compound. The modulus formalism equation is the inverse of complex dielectric permittivity as Equation (9) [40]:(9)$$M*(\omega )=\frac{1}{\varepsilon *(\omega )}\frac{\varepsilon^{\prime }}{{{(\varepsilon }^{\prime })}^{2}{+(\varepsilon }^{\prime \prime }{)}^{2}}+i\frac{\varepsilon^{\prime \prime }}{{{(Z}^{\prime })}^{2}{+}^{(}\varepsilon^{\prime \prime }{)}^{2}}{=M}^{\prime }+iM^{\prime \prime }$$

Figure 7a illustrates the real electrical modulus, *M*′ of Sr_2_CaTeO_6_ within frequencies of 50 Hz to 1 MHz. From this figure, there are two regions of variation, which are the low gradient plot at lower frequencies and the high gradient and dispersion plot at around 100 kHz and above. Figure 7b shows the imaginary electrical modulus, *M*″ of this compound. It can be observed that there is a presence of a peak at initial frequencies and at frequencies around 100 kHz. Figure 7c shows the Bode plot of imaginary impedance, *Z*″, at the same frequencies region. It can be seen that there are peaks present at frequencies lower than 50 Hz at each temperature. Figure 7d shows the capacitance values in this compound at each temperature. The temperature-dependent capacitive property can be observed from this plot. Figure 7e shows the relaxation times variation against the inverse of different temperatures graph. From this graph, the almost linear shape of variation can be seen across the temperature range.

From Figure 7a, the small values of *M*′ at initial frequencies can be related to the tendency for electrode effects to be suppressed and not contribute to the total electrical conductivity in this sample. In the low gradient region, it shows the almost plateau variation in *M*′ and low electrical stiffness. This variation indicates that long-range conduction exists within the compound. On the other hand, the dispersed high slope region indicates the presence of short-range electrical conductivity at high frequencies, which means the localised conduction mechanisms of charge carriers exist within this frequency range. It is believed that carriers only move within their respective potential wells, led by the absence of the restoring force [2]. With the presence of higher frequencies of an external field, the movement of charge carriers was merely conducted by the external field and hence, increased the electrical stiffness. The dispersion could also indicate the temperature-dependent relaxation process could take place within this frequency range. As the temperature elevated, the *M*′ values decreased, which can be deduced that motions of charge carriers enhanced with temperatures. The shifting of the dispersion region towards lower frequencies as temperature elevated could indicate the short-range conduction took place.

From Figure 7b, the primary peak indicates the presence of grain boundaries effect towards compound’s electrical characteristics. Nevertheless, the exact peak frequencies cannot be determined directly from this plot since lower frequencies are needed. The Bode plot from Figure 7c supports that the grain boundary effects took place at lower frequencies. However, the plot does not show any presence of grain effect. The grain effect can be studied by analysing the *M*″ plot. The peak at high frequencies in Figure 7b refers to as grain effect towards the electrical property. This peak correlates well with the dispersion region in *M*′ plot. As the temperature increased, the value of *M*″ increased with the *M″_max_* shifted towards the higher frequencies region; the shift supported *Tan δ* results to indicate the relaxation frequencies region. The difference in *M*″ values at each temperature suggests different capacitance at respective temperature and indicate a thermal effect on capacitance values. It can be seen that the values show a decrement from ambient temperature to 353 K. This can be understood since the dynamic of electrons is more progressive as temperature increases and hence, reducing the static charge storage and increasing the capacitive loss. This variation can be linked to dielectric real permittivity and dielectric loss in the sample shown in Figure 6a,b. Meanwhile, the shifting of *M″_max_* frequencies indicates the different relaxation times at respective temperatures by calculating the *ω*^−1^. From Figure 7e, the variation in relaxation peaks against the inverse in temperature followed the Arrhenius equation and could provide the activation energy of the relaxation process of electrical conductivity at specified frequencies. From this equation, the calculated activation energy, *E_a_,* obtained was 0.09 eV which was slightly different from DC conduction *E_a_*. The relaxation process from the *M”* in this sample can presumably be due to AC conductivity. Hence, the difference between *E_a_* of DC and AC conductivities could indicate that different charge carriers were involved in both conductivities. Shorter relaxation times with temperature increases implied that more kinetic energy conversion contributes toward the dynamic of the charge carriers.

In order to understand the complete electrical conductivity of a sample, AC conductivity characterisation is needed to understand the electrical conductivity that takes place in the materials with the presence of the external electric field. Figure 8a shows the AC conductivity of the Sr_2_CaTeO_6_ from the frequency of 50 Hz until 1 MHz. From this plot, there are regions of plateau and dispersion regions, separated by a solid blue line in the plot between 10 kHz and 100 kHz. The graph in Figure 8b was plotted in order to further understand the AC conduction mechanism in Sr_2_CaTeO_6_. This graph shows the variation in s against different temperatures. The value of s can be obtained from the slope of log σ_AC_ vs. log f. According to this plot, the variation in s increases from ambient temperature until 323 K, decreases at 333 K, and increases back until 353 K, with the s values lying within the range of 0.1 to 0.8.

From Figure 8a, in region 1, the plateau region could be related to the long-range DC conductivity at lower frequencies, while the dispersion region could be related to localised AC conductivity at higher frequencies. The increment of electrical conductivity from ambient temperature to 353 K correlates with the NTCR behaviour in grain boundaries and grain structures. The shape of this plot can be explained by the Jonscher Equation (10) [40]:(10)$$\sigma_{T}{=\sigma }_{DC}{+A\omega }^{s}$$
where *A* = pre-exponential constant, which indicates the polarisation strength, while *s* = power-law exponent, which indicates the degree of carriers’ interactions with lattices structure. In the Jonscher equation, *σ_DC_* is related to DC conductivity, while *Aω^s^* is related to AC conductivity.

Mechanisms that are commonly utilised to elucidate the AC conductivity properties of perovskites related to the variation in *s* at different temperatures. For instance, variable range hopping (VRH), small polaron hopping (SPH), quantum mechanical tunnelling (QMT), non-overlapping small polaron tunnelling (NSPT), and overlapping large polaron tunnelling (OLPT) mechanisms [18,41,42,43,44,45,46]. According to the pattern in Figure 8b, a variation in *s* values from 298 K to 323 K could be linked to NSPT or SPH. However, NSPT could be the most suited model since the *E_a_* from *M*″, which were due to AC conductivity, is comparable to other double perovskites, as reported before [43]. Furthermore, the variation in *s* values from 323 K to 353K is in accordance with the variation in the OLPT model.

The AC conductivity of the NSPT model is given by the Equation (11) [43]:(11)$$\sigma^{\prime }\;\left(\omega \right)=\frac{\pi^{2}e^{2}{\;k}_{B}{T\alpha }^{-1}{[N\left(E_{F}\right)]}^{2}{\omega R}_{\omega }{}^{4}}{12}$$
where $N\left(E_{F}\right)$ = the density of the localised state, $k_{B}$ represents Boltzmann’s constant, T = the absolute temperature in K, and $R_{\omega }$ = the tunnelling length at frequency ω.

The $R_{\omega }$ can be elaborated as in Equation (12) [43]:(12)$$R_{\omega }=\frac{1}{2\alpha }[\;Ln\;\left(\frac{1}{{\omega \tau }_{o}}\right)-\frac{W_{m}}{k_{B}T}]$$
where *α*^−1^ = spatial extension of the polaron, *τ_o_* = relaxation times in s, and *W_m_* = polaron hopping energy in eV.

The frequency exponent, *s*, predicted by this model is given by the Expression (13) [43]:(13)$$s=1-\frac{{4k}_{B}T}{W_{m}{-k}_{B}{TLn(\omega \tau }_{o})}$$
whereas the AC conductivity for the OLPT model is given by Equation (14) [46,47]:(14)$$\sigma^{\prime }\left(\omega \right)=\frac{\pi^{2}}{12}\;\frac{e^{2}{{(k}_{B}T)}^{2}{[N\left(E_{F}\right)]}^{2}{\omega R}_{\omega }{}^{4}}{{2\alpha k}_{B}T+\frac{W_{m}r_{p}}{r_{\omega }^{2}}}$$
where α = decay parameter for the localised wave function and $r_{p}$ = the polaron radius.

The frequency exponent, *s*, predicted by this model is given by the Expression (15) [46]:(15)s=1−8αRω+6WmrpRωkBT[(2αRω+Wmrp)RωkBT]2

Charge carriers in the sample probably originated from electrons due to oxygen vacancies, which transform into polarons due to their movement in the deformed lattice of the sample when induced polarisation follows the electrons. In the case of small polarons, the size of the polarons was smaller than the lattice constant of the compound. Meanwhile, the size of polarons was greater than the lattice constant in large polarons formations. The large polarons that formed with shallow bound states were attributed to Coulomb interaction between charge carriers and polarisable lattice. Both types of charge carriers were able to move across sites with potential energy well by tunnelling mechanisms between oxygen vacancies [48]. In both NSPT and OLPT models, the polaron wells overlapped at two sites and hence, reducing the polaron energy to travel across the sites.

Towards understanding the basic optical property of this material, it is important to analyse the UV–vis characterisation study. Figure 9a exhibits the UV–vis absorption plot of Sr_2_CaTeO_6_. The plot was calculated from Kubelka–Munk equation as in Equation (16) [47]:(16)$$F(R)=\frac{{\left(1-R\right)}^{2}}{2R}$$
where *R* = diffuse reflectance of compounds. Figure 9b,c shows the Tauc plots that utilised Equation (17):(17)$${\left[F\left(R\right)hv\right]}^{1/n}=A\left({hv-E}_{opt}\right)$$
where *hv* = photon energy, *A* = proportional constant, and *E_opt_* = energy of optical bandgap. Meanwhile, the values of *n* can vary based on the transition type in a material, where *n* = 1/2 indicates the direct allowed transition, *n* = 2 indicates the indirect allowed transition, *n* = 3/2 indicates the direct forbidden transition, and *n* = 3 indicates the indirect forbidden transition [47].

From Figure 9a, the value of the absorption point is around 343.2 nm. Hence, the calculated value of *E_opt_* from this absorption plot is 3.61 eV. Meanwhile, Figure 9b utilised the indirect allowed transition of electrons. From this figure, the extrapolated value of *E_opt_* is 2.92 eV. From Figure 9c, the fitting and extrapolation of *E_opt_* applied the direct allowed transition of electrons. The value of *E_opt_* measured from this transition is 3.75 eV, which showed much larger energy needed for the transition of electrons from the highest valence band to the lowest conduction band, despite only photons being involved in this kind of transition.

The indirect allowed type of transition was considered because of the presence of the hump in the absorption plot before the values become quenched at a higher wavelength. This could indicate that the indirect allowed transition was involved in this compound with the inclusion of photons and phonons movements. The obtained value of *E_opt_* was comparable to the result from other reports [41]. However, based on the Kubelka–Munk absorption wavelength, the direct allowed *E_opt_* can be favoured as well as *E_opt_* values from both Kubelka–Munk and allowed direct transition is comparable. Furthermore, the shape of the plots is more similar despite their shape being inverse to each other. Values from all plots indicate the semiconductor range of *E_opt_* in this compound. However, the *E_opt_* values obtained from direct allowed transition lie in a wide semiconductor band gap range. Compared to other tellurium-based double perovskites, the *E_opt_* of allowed indirect transition in this compound showed a much smaller value [37,47] despite more distortion within this compound. This smaller *E_opt_* could be linked to the crystallite size difference between the compounds nonetheless [49]. Additionally, the smaller *E_opt_* in this compound could be related to possible minor oxygen vacancies existence that could involve in the formation of the energy level of impurities within the sample band gap or lifted the Fermi energy and thus, reducing the *E_opt_* value [50]. However, as per refinement analysis, the basic lattice structure of the compound was well-formed. Furthermore, the caution measure was taken by using a slow cooling rate to maintain the stoichiometry of the compound. Hence, it can be deduced that the oxygen vacancies formed are very minimal and should not affect the *E_opt_* majorly. The absorbance spectra of Sr_2_CaTeO_6_ and its semiconducting range of *E_opt_* indicate its potential to be utilised in UV range applications such as ultraviolet detector devices.

## 4. Conclusions

Sr_2_CaTeO_6_ was successfully synthesised by using a conventional solid-state procedure. The formed compound was found to crystallise in monoclinic *P2_1_/n* symmetry by refinement method with the formation of B–O bonds confirmed through infrared spectrum study. Good characteristics of dielectric in this compound were exhibited with ~10,000 and 0.25 value range of dielectric real permittivity and loss, respectively, at the low frequencies in room temperature. These values were comparable and even larger than some reported tellurium-based double perovskites in the *AA’BB’O_6_* configuration. The relatively small dielectric loss in this compound could be affected by electrical conductions due to vacancy defects or other factors, as indicated through impedance and modulus paths. The semiconducting behaviour of the grain was supported by the optical band gap value by direct or indirect allowed transition plots.

## Figures and Tables

**Figure 1 materials-15-04363-f001:**
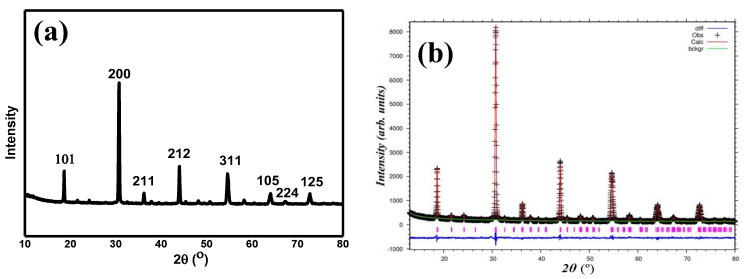
(**a**) XRD graph and (**b**) refinement of XRD data of Sr_2_CaTeO_6_. The black lines, pink lines, and blue lines are the observed patterns, the calculated data, and the difference, respectively.

**Figure 2 materials-15-04363-f002:**
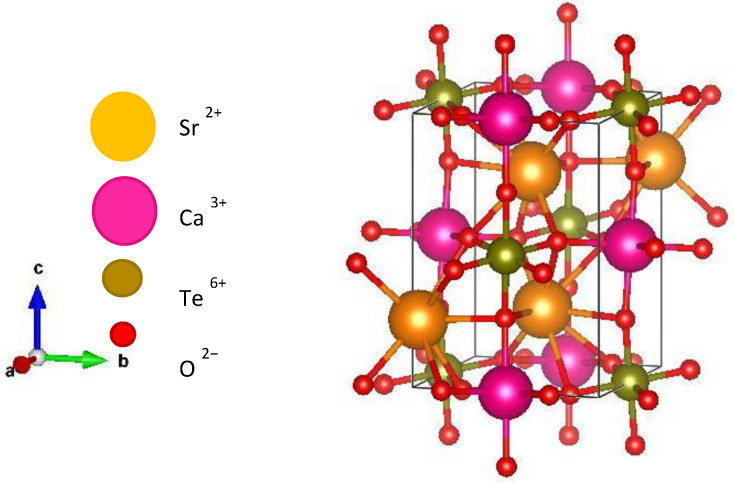
Image of refined XRD in Sr_2_CaTeO_6_ from *bc* plane.

**Figure 3 materials-15-04363-f003:**
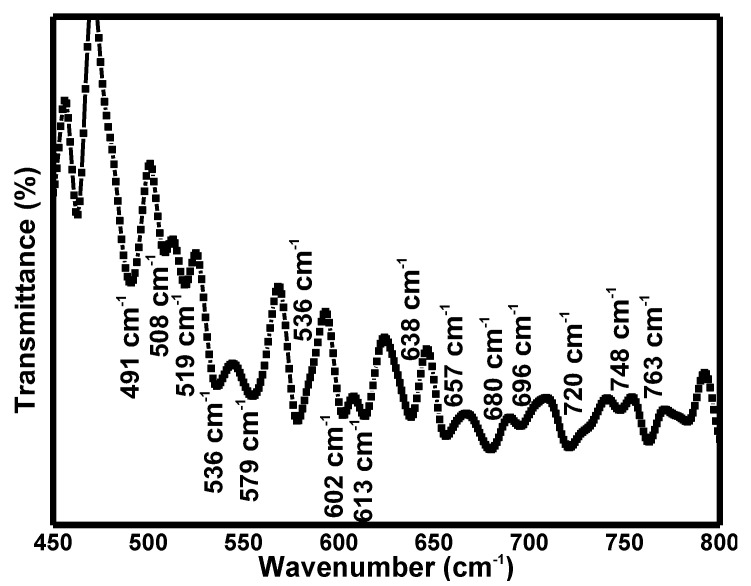
FTIR spectrum of Sr_2_CaTeO_6_.

**Figure 4 materials-15-04363-f004:**
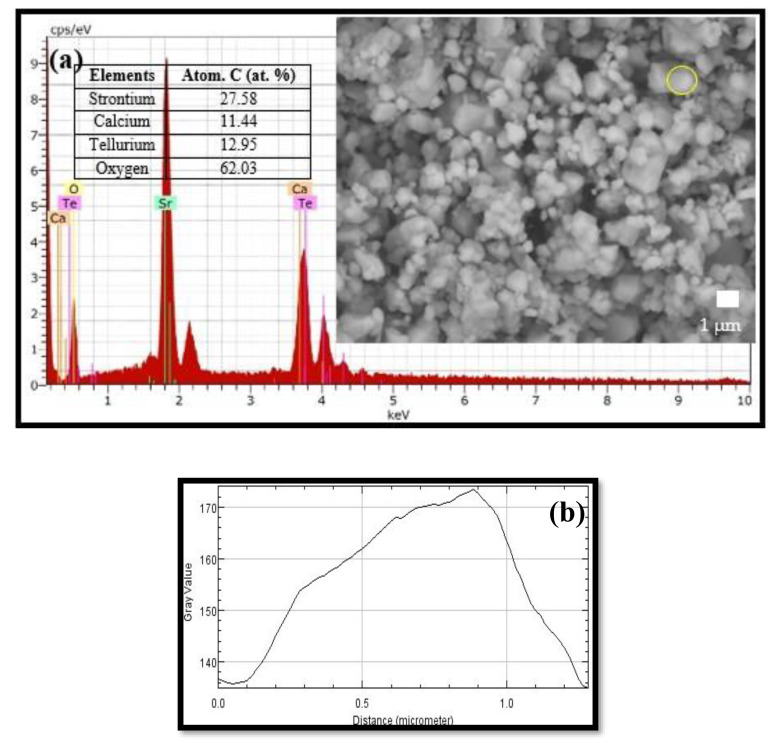
(**a**) SEM images (10 K magnification) and EDX plot, (**b**) illustration of the top half of grain, and (**c**) grain size distributions of Sr_2_CaTeO_6_.

**Figure 5 materials-15-04363-f005:**
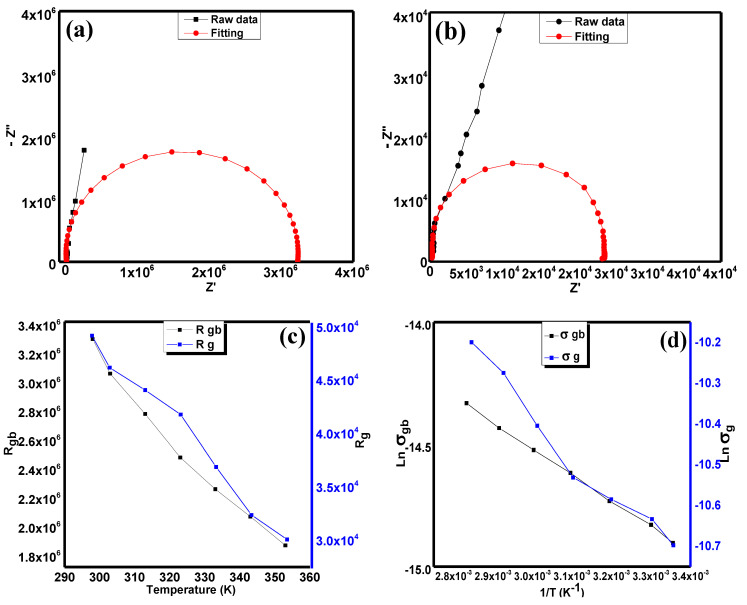
Nyquist plot fitting at (**a**) full frequencies range and (**b**) at high frequencies, (**c**) grain boundaries and grain resistance vs. temperature, and (**d**) ln DC conductivity vs. inverse temperature for grain boundaries and grain structure of Sr_2_CaTeO_6_.

**Figure 6 materials-15-04363-f006:**
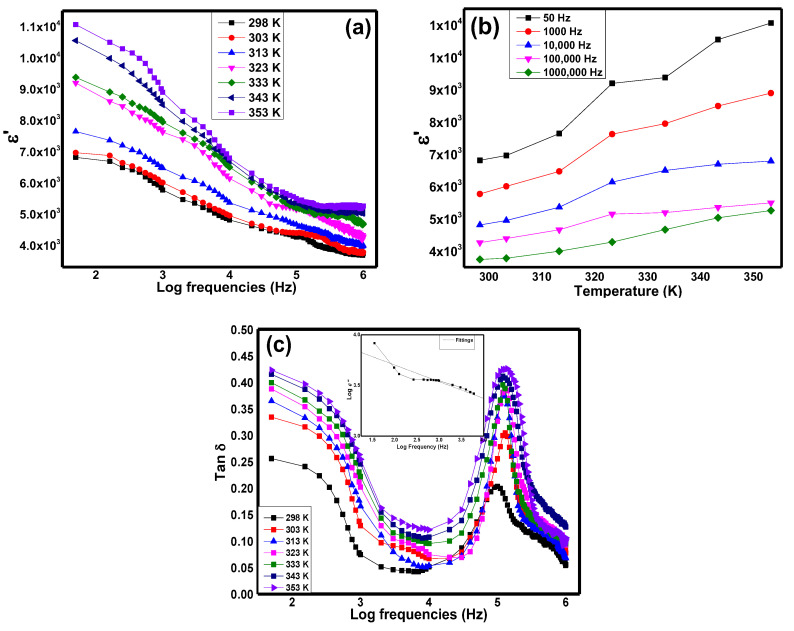
Variation in *ε*’ with (**a**) frequencies, (**b**) temperature, and (**c**) variation in *Tan δ* with frequencies of Sr_2_CaTeO_6_.

**Figure 7 materials-15-04363-f007:**
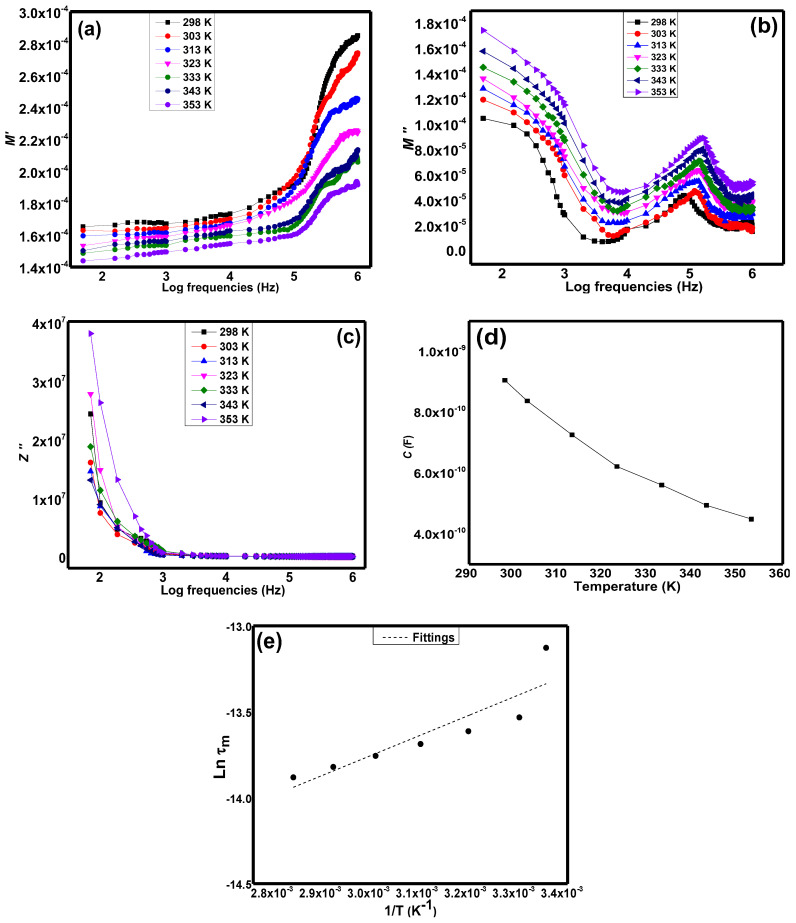
(**a**) The real, (**b**) imaginary electrical modulus, (**c**) reactance impedance against different frequency logarithmic, (**d**) capacitance values at various temperatures, and (**e**) ln relaxation times vs. inverse in temperature of Sr_2_CaTeO_6_.

**Figure 8 materials-15-04363-f008:**
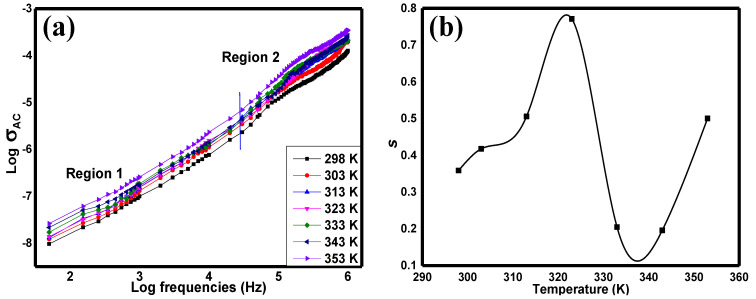
(**a**) AC conductivity plot against frequencies and (**b**) power-law exponent variation with various temperatures.

**Figure 9 materials-15-04363-f009:**
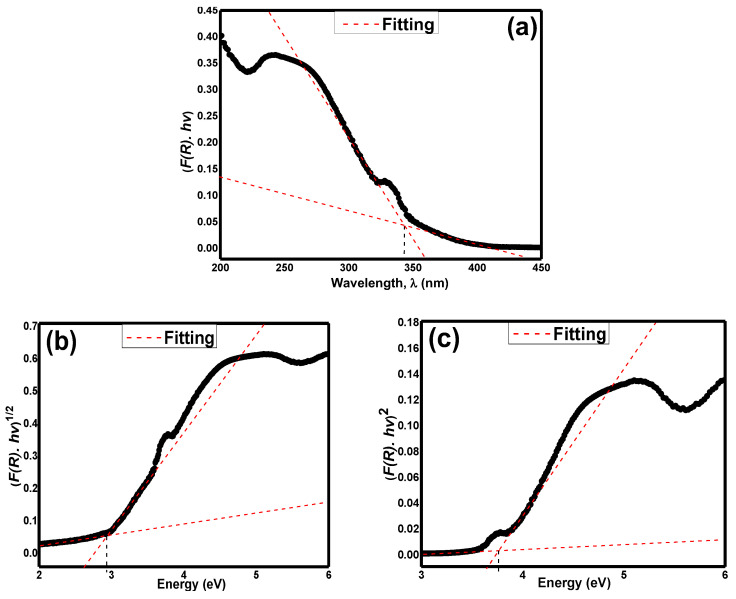
(**a**) The Kubelka–Munk, Tauc plotting of (**b**) allowed indirect transition and (**c**) allowed direct transition in Sr_2_CaTeO_6_ (dotted lines are the fitting for optical band gap).

## Data Availability

Not applicable.

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
