# Peer review of "Dielectric, AC Conductivity, and DC Conductivity Behaviours of Sr2CaTeO6 Double Perovskite"

_materials, 2022, doi:10.3390/ma15124363_

Round 1

Reviewer 1 Report

The authors synthesize Sr2CaTeO6 materials and characterized their properties, including crystallographic structure, electric conductivity, bandgap etc.  The work is publishable on the journal after considering the below comments.

1) Previous research on Se2CaTeO6 is not listed well in Introduction.

2) Page 180: what's CN:12?  What is CN:6?

3) X-ray refinement indicated the crystalline size is 10.71 nm (Fig1, Table 1, and Line 191) while the SEM image indicated the grain size is 0.7-2.5um (Fig 4, Line 226-231).  Please explain why the difference between the two conclusion.

4) What's the element ratio measured from the EDX spectrum (Fig 4a)?

5) Line 232, "EDS graph in Figure 4(c)", should be Figure 4(a).

6) Caption of Figure 4 (a) "5K and 10K magnification": only one magnification in Fig4 a.  It is better to draw a scale bar on the SEM image. 

7) Fig 4c: the grain size is much larger than 10.71 nm measured from X-ray diffraction.  Please explain.

8) It is not clear how to get resistance of grain boundaries.

9) 8) It is not clear how to get resistance of grains.

10) The authors contribute the high conductivity and semiconductor behaviors to oxygen vacancies.  Please provide oxygen occupancy in the material, which is fitted from X-ray refinement, to support the conclusion.  

11) Typo in Fig 3: 509 should be 508.

12) Table 3: please list the dimensions of samples when discuss capacitance (C) and resistances (Rgb and Rg).  The values of conductivities are not match the data in Fig. 5d.

13) Please ref one paper (Patrycja Makuła, Michał Pacia, and Wojciech Macyk

The Journal of Physical Chemistry Letters 2018 9 (23), 6814-6817;  How To Correctly Determine the Band Gap Energy of Modified Semiconductor Photocatalysts Based on UV–Vis Spectra.  DOI: 10.1021/acs.jpclett.8b02892) when calculate bandgaps from UV-spectra in Fig 9. 14) Some Tables, such as Table 3 and Table 4, are not necessary as the data in Tables is plotted or described in main text.  

Author Response

Dear editor/reviewer,

Thank you very much.

Reviewer 2 Report

The authors studied preparation and characterization of a double perovskite material, Sr2CaTeO6. The characterization included structural, optical, dielectric and electrical measurements. I notice that this paper is important especially in the basis on academic viewpoint. Overall, the manuscript is written in clear English. The paper may be published, however, after some minor corrections, as justified in the following points:

1) Figures 5(a) and 5(b): nothing was mentioned about how the Nyquist fittings were obtained. Din the authors employed a common software for fittings?

2. Figure 8(a): Please reduce the size of solid symbols in order to follow the tendency of ac-conductivity as function of frequency. For a better visualization, I recommend the authors to plot the experimental points in a double logarithmic scale.

Author Response

(The authors gave the same response as above.)

Reviewer 3 Report

The paper shows results about dielectric, AC conductivity, and DC conductivity with Sr2CaTeO6 double perovskite. Authors analyzed structural, dielectric, and optical characteristics of Sr2CaTeO6 double perovskite. They made a discussion on its optical and dielectric properties. High electrical permittivity, low dielectric loss, and good capacitance over a range of temperatures possessed by this compound as shown in dielectric and electrical modulus studies indicated good potential values for capacitor applications. Its relatively high DC conductivity in grain at high frequencies and its increasing value with the temperature are typical of a semiconductor behavior. AC conductivity of this compound was found to contribute to the dielectric loss in grain structure.

Dear authors. Thank you very much for your value paper dielectric, AC conductivity, and DC conductivity with Sr2CaTeO6 double perovskite. I have some comments and suggestions, what should be considered in presented to review paper.

Comments and suggestions:

1. Introduction well describes the main information about perovskite materials. Authors present magnetic properties of the materials. They indicate also dielectric properties, such as DC and AC electrical conductivity, electrical permittivity and losses factor.

2. Next chapter shows how the material was prepared, which was used in experiments. I think, the explanation is very detail, what means correct, because the material preparation is important part of the experiment. It let to repeat this experiment for other researchers.

3. please use full name of permittivity and conductivity, what means electrical permittivity and electrical conductivity. There are some kinds of conductivity, such as electrical, thermal etc.

4. Formulas should be written using Word standard – Equations, please correct. Presented formulas are not very complicated from editor point of view.

5. Parameters in mentioned formulas should be explained using also their units, please complete.

6. Fig.1 – graph, please describe the axis X and Y, and add please unit of the properties of the axis.

7. When authors present some formulas, please indicate the source – references, you used.

8. Fig.6 – IMPORTANT – please explain obtained distributions of conductivity, tan(delta) and electrical permittivity as a function of frequency and temperature form physical point of view.

9. Talking about AC and DC, authors should use capital letter, not ac or dc.

Author Response

(The authors gave the same response as above.)
